# Revisiting Bisimulation: A Sampling-Based State Similarity Pseudo-metric

**Charline Le Lan**
University of Oxford
charline.lelan@stats.ox.ac.uk

**Rishabh Agarwal**
Google DeepMind
rishabhagarwal@google.com

## 1 Introduction

In reinforcement learning (RL), we typically deal with systems with large or continuous states encoded in an unstructured way. Because it is not possible to represent the value of each state, it is necessary to learn a structured representation from limited state samples to express the value function in a more meaningful way. One approach to do so is to endow the set of states with a behavioral metric, such that two states that are close in the metric space are also close in the space of value functions. While there exists some notions of state similarity, they are either not amenable to sample-based algorithms (Ferns et al., 2004; 2005), need additional assumptions (Castro, 2020; Zhang et al., 2020; Agarwal et al., 2021) or yield limited theoretical guarantees (Castro et al., 2021). In this paper, we present a new behavioural pseudo-metric, PMiCo, to overcome these shortcomings. PMiCo is based on a recent sampling-based behavioural distance, MICo (Matching under Independent Couplings; Castro et al., 2021), but enjoys more interesting theoretical properties, which we also illustrate empirically.

## 2 Background

**Reinforcement learning.** We consider a Markov decision process (MDP) $\mathcal{M} = \langle \mathcal{X}, \mathcal{A}, r, \mathcal{P}, \gamma \rangle$ (Puterman, 1994) with finite state space $\mathcal{X}$, finite action space $\mathcal{A}$, transition dynamics $\mathcal{P} : \mathcal{X} \times \mathcal{A} \to \mathscr{P}(\mathcal{X})$, reward function $r : \mathcal{X} \times \mathcal{A} \to \mathbb{R}$, and discount factor $\gamma \in [0, 1)$. A stationary policy $\pi : \mathcal{X} \to \mathscr{P}(\mathcal{A})$ is a mapping from states to distributions over actions. We denote the set of all policies by $\Pi$. For any policy $\pi \in \Pi$, the value function $V^\pi(x)$ measures the expected discounted sum of rewards received when starting from state $x \in \mathcal{X}$ and acting according to $\pi$ and satisfies Bellman's equation $V^\pi(x) := \mathbb{E}_{a \sim \pi(x)}\left[r_x^a + \gamma \mathbb{E}_{x' \sim P_x^a}\left[V^\pi(x')\right]\right]$. In RL, the goal is to find the optimal policy $\pi^* = \mathrm{argmax}_{\pi' \in \Pi} V^{\pi'}(x)$ for all states $x \in \mathcal{X}$ from *sample interactions* with the environment, which are a sequence of states, actions and rewards $(X_t, A_t, R_t)_{t \geq 0}$.

**Bisimulation.** The *bisimulation metric* (Ferns et al., 2004) describes two states as similar if their immediate rewards are close and they transition to next states which are also considered similar. More formally, denoting $\mathcal{M}(\mathcal{X}) = \{d \in [0, \infty)^{\mathcal{X} \times \mathcal{X}} \mid \forall x, y, z \in \mathcal{X}, d(x,x) = 0; d(x,y) = d(y,x); d(x,y) \leq d(x,z) + d(z,y)\}$ the set of *pseudo-metrics* [1] on $\mathcal{X}$, the bisimulation metric $d^\sim$ is defined as the unique fixed-point (unicity resulting from Banach Fixed point theorem, see e.g. Ferns et al., 2005) of the operator $T_W : \mathcal{M}(\mathcal{X}) \to \mathcal{M}(\mathcal{X})$, that is the metric $d \in \mathcal{M}(\mathcal{X})$ for which $T_W(d) = d$, where $T_W(d)(x,y) = \max_{a \in \mathcal{A}}\left[|r(x,a) - r(y,a)| + \gamma W_d\left(P(x,a), P(y,a)\right)\right]$. Here, $W_d$ is the 1-Wasserstein metric (Villani, 2008) and describes the minimal cost of transporting probability mass from $\mu \in \mathscr{P}(\mathcal{X})$ to $\nu \in \mathscr{P}(\mathcal{X})$. Castro (2020) extended the canonical bisimulation to an on-policy counterpart. The $\pi$-*bisimulation* $d_\sim^\pi$ is the fixed point of the operator $T_W^\pi(d)(x,y) = |r_x^\pi - r_y^\pi| + \gamma W_d\left(P_x^\pi, P_y^\pi\right)$ with $r_x^\pi = \sum_{a \in \mathcal{A}} \pi(a \mid x) r_x^a$, $P_x^\pi = \sum_{a \in \mathcal{A}} \pi(a \mid x) P(x,a)(\cdot)$ for all $x \in \mathcal{X}$. $V^\pi$ (resp. $V^*$) is Lipschitz continuous with respect to $d_\sim^\pi$ (resp. $d^\sim$), that is $|V^\pi(x) - V^\pi(y)| \leq d_\sim^\pi(x,y)$ for any $x, y \in \mathcal{X}$ and any $\pi \in \Pi$. However, this metric is computationally expensive and introduces a bias under sampled transitions (Ferns et al., 2006; Comanici et al., 2012).

To overcome these issues, Castro et al. (2021) introduced the concept of *MICo distance* $U^\pi$ as the fixed point of the operator $T_M^\pi : \mathbb{R}^{\mathcal{X} \times \mathcal{X}} \to \mathbb{R}^{\mathcal{X} \times \mathcal{X}}$ where $(T_M^\pi U)(x,y) = |r_x^\pi - r_y^\pi| + \gamma \mathbb{E}_{x' \sim P_x^\pi, y' \sim P_y^\pi}[U(x', y')]$ for all $U : \mathcal{X} \times \mathcal{X} \to \mathbb{R}$. While it can be learnt from samples, the MICo operator **does not** define a pseudo-metric as there exists states $x \in \mathcal{X}$ such that $d(x,x) \neq 0$.

---

[1] formally, a metric is a pseudo-metric satisfying the additional condition $d(x,y) = 0 \iff x = y$

| DISTANCE | SAMPLE-BASED | PSEUDO-METRIC | $V^\pi$-LIPSCHITZ CONTINUITY |
|---|---|---|---|
| $\pi$-BISIMULATION | ✗ | ✓ | ✓ |
| MICo | ✓ | ✗ | ✓ |
| PROJECTED MICo | ✓ | ✗ | ✗ |
| PMICo | ✓ | ✓ | ✓ |

Table 1: Categorization of behavioural distances according to their sample-based definition, pseudo-metric properties and continuity implications.

Since most algorithmic methods used to measure distances need the property of zero self-distance, Castro et al. (2021) rely in practice on the projection $\Pi U^\pi$ of the MICo distance, which is algorithmicly more complex and does not guarantee the same upper bound on the value function as MICo .

## 3 THE PMICo PSEUDO-METRIC

As an alternative to the above distances, we introduce a new (pseudo) metric, inspired by the MICo distance, which we call PMiCo (**P**seudo-metric **M**atching under **i**ndependent **C**ouplings). This metric is desirable because it can be learnt from transition samples and yields strong theoretical guarantees. Given $\pi \in \Pi$, the PMiCo update operator $T_P^\pi : \mathbb{R}^{\mathcal{X} \times \mathcal{X}} \to \mathbb{R}^{\mathcal{X} \times \mathcal{X}}$ is

$$(T_P^\pi U)(x, y) = |r^\pi(x) - r^\pi(y)| + \mathbf{1}_{[x \neq y]} \gamma \mathbb{E}_{x' \sim P_x^\pi, y' \sim P_y^\pi}[U(x', y')] \tag{1}$$

where $\mathbf{1}$ is the indicator function. The PMiCo distance between two distinct points is always positive, making it an appealing candidate for function approximation, without the need of a projection.

**Theorem 1.** *The PMiCo operator $T_P^\pi$ is a contraction mapping on $\mathbb{R}^{\mathcal{X} \times \mathcal{X}}$ w.r.t. $L^\infty$ norm and its fixed point $U_P^\pi$ is a pseudo-metric.*

One question that naturally arises is what are the guarantees of a representation induced by the PMiCo metric. Similarly to the MICo distance, we have the following on-policy guarantee for PMiCo.

**Theorem 2.** *For any policy $\pi \in \Pi$ and states $x, y \in \mathcal{X}$, $|V^\pi(x) - V^\pi(y)| \leq U_P^\pi(x, y) \leq U^\pi(x, y)$.*

We provide all proofs in Appendix A and summarize the properties of the various distances in Table 1.

We now conduct an empirical evaluation to illustrate the correctness of Theorem 2. Following Castro et al. (2021), we use *Garnet MDPs* (Archibald et al., 1995; Piot et al., 2014). For a fixed number of states and actions, we sample 100 stochastic policies $\{\pi_i\}$ and measure the average signed gap $\frac{1}{100|\mathcal{X}|^2} \sum_i \sum_{x,y} d(x, y) - |V^{\pi_i}(x) - V^{\pi_i}(y)|$. We consider four distances: the PMiCo metric, the MICo and projected MICo distances and the $\pi$-bisimulation. Figure 1 clearly shows that the PMiCo metric yields a tighter bound than its sample-based counterpart, the MICo distance, on average. While the bound from the $\pi$-bisimulation and projected MICo distances is tighter, unlike PMiCo, the $\pi$-bisimulation is biased when approximated using only sampled transitions and the projected MICo can be negative and may not upper bound the value differences.

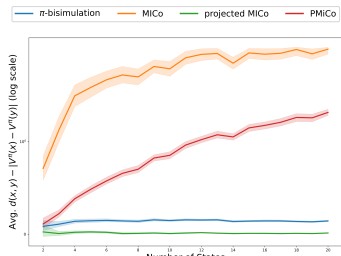

Figure 1: Gap between the difference in values and various distances for Garnet MDPs with 3 actions. Shaded areas represent 95 % confidence intervals.

## 4 DISCUSSION

In this paper, we presented PMiCo, a novel sample-based pseudo-metric. We saw it supports the Lipschitz continuity of the value function and induces a more coarse representation on the state space than the previously introduced MICo distance (Castro et al., 2021), suggesting it can lead to better generalization (Le Lan et al., 2021). Our experiments on Garnet MDPs show the tighter bound yielded by the PMiCo metric. In deep RL, behavioural metrics are often incorporated into the learning process as auxiliary tasks, to help shape the representation learnt by the agent (Gelada et al., 2019; Agarwal et al., 2021; Zhang et al., 2020; Castro et al., 2021). As such, training a network on auxiliary predictions such that the distance between two latent states corresponds to PMiCo could be a simple and effective auxiliary task for deep RL.

ACKNOWLEDGEMENTS

The authors would like to thank Pablo Samuel Castro and the anonymous reviewers for detailed feed-back on this manuscript. We would also like to thank the Python community (Van Rossum & Drake Jr, 1995; Oliphant, 2007) for developing tools that enabled this work, including NumPy (Oliphant, 2006; Walt et al., 2011; Harris et al., 2020) and Matplotlib (Hunter, 2007).

URM STATEMENT

Charline Le Lan meets the URM criteria of ICLR 2023 Tiny Papers Track.

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

## A    Proofs for Section 3

**Lemma 1.** *The PMiCo operator $T_P^\pi$ is a contraction mapping on $\mathbb{R}^{\mathcal{X} \times \mathcal{X}}$ with respect to the $L^\infty$ norm.*

*Proof.* Let $U, U' \in \mathbb{R}^{\mathcal{X} \times \mathcal{X}}$. Then, by definition of the PMiCo pseudo-metric, it follows that for any $x, y \in \mathcal{X}$,

$$|(T_P^\pi U)(x,y) - (T_P^\pi U')(x,y)| = \left| \gamma \mathbf{1}_{[x \neq y]} \mathbb{E}_{x' \sim P_x^\pi, y' \sim P_y^\pi} (U - U')(x',y') \right| \leq \gamma \|U - U'\|_\infty .$$

$\square$

**Theorem 1.** *The PMiCo operator $T_P^\pi$ is a contraction mapping on $\mathbb{R}^{\mathcal{X} \times \mathcal{X}}$ w.r.t. $L^\infty$ norm and its fixed point $U_P^\pi$ is a pseudo-metric.*

*Proof.* The first part of Theorem 1 follows from Lemma 1.

By application of Banach's fixed-point theorem and by the completeness of $\mathbb{R}^{\mathcal{X} \times \mathcal{X}}$ under the $L^\infty$ norm, it follows that the PMiCo operator has a unique fixed point $U_P^\pi \in \mathbb{R}^{\mathcal{X} \times \mathcal{X}}$ and applying $T_P^\pi$ repeatedly to an initial function $U \in \mathbb{R}^{\mathcal{X} \times \mathcal{X}}$ converges to $U_P^\pi$.

It it easy to see that $U_P^\pi$ is symmetric, non-negative and has zero self-distance ($\forall x \in \mathcal{X}, U_P^\pi(x,x) = 0$).

To prove the triangle inequality, we adapt the proof of proposition 4.10 from Castro et al. (2021) and rely on a proof by induction.

We define a sequence of iterates $(U_k)_{k \geq 0}$ in $\mathbb{R}^{\mathcal{X} \times \mathcal{X}}$ by $U_0(x,y) = 0$ for all $x, y \in \mathcal{X}$, and $U_{k+1} = T_M^\pi U_k$ for each $k \geq 0$.

The base case of the inductive argument is clear from the choice of $U_0$.
For the inductive step, assume that for some $k \geq 0$, $U_k(x,y) \leq U_k(x,z) + U_k(z,y)$ for all $x, y, z \in \mathcal{X}$. For $x, y, z \in \mathcal{X}$,
**Case I**. If $z = x$ or $z = y$, we have $U_{k+1}(x,y) = U_{k+1}(x,z) + U_{k+1}(z,y)$ by zero self-distance argument.
**Case II**. If $x = y$, we have $U_{k+1}(x,y) = 0 \leq U_{k+1}(x,z) + U_{k+1}(z,y)$ by non-negativity of $U_{k+1}$.
**Case III**. If $x \neq y \neq z$, we have

$$
\begin{aligned}
U_{k+1}(x,y) &= |r^\pi(x) - r^\pi(y)| + \gamma \mathbb{E}_{x' \sim P_x^\pi, y' \sim P_y^\pi}[U_k(x',y')] \qquad \text{because } x \neq y \\
&\leq |r^\pi(x) - r^\pi(z)| + |r^\pi(z) - r^\pi(y)| + \gamma \mathbb{E}_{x' \sim P_x^\pi, y' \sim P_y^\pi, z' \sim P_z^\pi}[U_k(x',z') + U_k(z',y')] \\
&= \left( |r^\pi(x) - r^\pi(z)| + \gamma \mathbb{E}_{x' \sim P_x^\pi, z' \sim P_z^\pi}[U_k(x',z')] \right) + \Big( |r^\pi(z) - r^\pi(y)| \\
&\quad + \gamma \mathbb{E}_{z' \sim P_z^\pi, y' \sim P_y^\pi}[U_k(z',y')] \Big) \\
&= U_{k+1}(x,z) + U_{k+1}(z,y) \qquad \text{because } x \neq z, z \neq y
\end{aligned}
$$

as required. Hence, for any $k \geq 0$, $U_{k+1}$ satisfy the triangle inequality. $U_k \to U_P^\pi$ (using lemma 1), so by taking limits on either side of the inequality, we recover that $U_P^\pi$ itself satisfies the triangle inequality. $\square$

**Lemma 2.** *For any policy $\pi \in \mathscr{P}(\mathcal{A})^{\mathcal{X}}$ and states $x, y \in \mathcal{X}$, we have $|V^\pi(x) - V^\pi(y)| \leq U_P^\pi(x,y)$.*

*Proof.* We adapt the proof of proposition 4.8 from Castro et al. (2021) and apply a coinductive argument (Kozen, 2006) to show that if

$$|V^\pi(x) - V^\pi(y)| \leq U(x,y) \text{ for all } x, y \in \mathcal{X} , \tag{2}$$

for some $U \in \mathbb{R}^{\mathcal{X} \times \mathcal{X}}$ symmetric in its two arguments, then we also have

$$|V^\pi(x) - V^\pi(y)| \leq (T_M^\pi U)(x,y) \text{ for all } x, y \in \mathcal{X} .$$

Since the hypothesis holds for the constant function

$$U(x,y) = \begin{cases} 2R_{\max}/(1-\gamma), & \text{for } x \neq y \\ 0, & \text{for } x = y \end{cases} \tag{3}$$

and $T_P^\pi$ contracts around $U_P^\pi$, the conclusion then follows.

**Case I**. If $x = y$, $V^\pi(x) - V^\pi(y) = 0 = (T_P^\pi U)(x,y)$.

**Case II**. If $x \neq y$, suppose Equation 2 holds. Then we have

$$V^\pi(x) - V^\pi(y) = r_x^\pi - r_y^\pi + \gamma \sum_{x' \in \mathcal{X}} P_x^\pi(x')V(x') - \gamma \sum_{y' \in \mathcal{X}} P_y^\pi(y')V(y')$$

$$\leq |r_x^\pi - r_y^\pi| + \gamma \sum_{x',y' \in \mathcal{X}} P_x^\pi(x')P_y^\pi(y')(V^\pi(x') - V^\pi(y'))$$

$$\leq |r_x^\pi - r_y^\pi| + \gamma \sum_{x',y' \in \mathcal{X}} P_x^\pi(x')P_y^\pi(y')U(x',y') \quad \text{by the inductive hypothesis}$$

$$= |r_x^\pi - r_y^\pi| + \mathbf{1}_{[x \neq y]}\mathbb{E}_{x' \sim P_x^\pi, y' \sim P_y^\pi}U(x',y') \qquad \text{because } x \neq y$$

$$= (T_P^\pi U)(x,y).$$

By symmetry, $V^\pi(y) - V^\pi(x) \leq (T_P^\pi U)(x,y)$, as required. $\qquad\square$

**Theorem 2.** *For any policy $\pi \in \Pi$ and states $x, y \in \mathcal{X}$, $|V^\pi(x) - V^\pi(y)| \leq U_P^\pi(x,y) \leq U^\pi(x,y)$.*

*Proof.* The left hand side of the inequality follows by application of lemma 2.

To prove the right hand side, we use a proof by induction. To do so, we define a sequence of iterates $(U_k)_{k \geq 0}$ and $(O_k)_{k \geq 0}$ in $\mathbb{R}^{\mathcal{X} \times \mathcal{X}}$ by $U_0(x,y) = O_0(x,y) = 0$ for all $x, y \in \mathcal{X}$, and $U_{k+1} = T_P^\pi U_k$ and $O_{k+1} = T_M^\pi O_k$ for each $k \geq 0$.

For $k = 0$, $U_0(x,y) - O_0(x,y) \leq 0$.

Assume that there exists $k > 0$ such that $U_k(x,y) \leq O_k(x,y)$. Then for any $x, y \in \mathcal{X}$

$$U_{k+1}(x,y) - O_{k+1}(x,y) = \gamma \mathbb{E}_{x' \sim P_x^\pi, y' \sim P_y^\pi}[\mathbf{1}_{[x \neq y]}[U_k(x',y') - O_k(x',y')]$$

$$\leq \mathbb{E}_{x' \sim P_x^\pi, y' \sim P_y^\pi}[U_k(x',y') - O_k(x',y')] \qquad \text{because } \mathbf{1}_{[x \neq y]} \leq 1$$

$$\leq 0 \qquad \text{because } U_k(x',y') - O_k(x,y) \leq 0$$

$T_P^\pi$ ( resp. $T_M^\pi$) are contraction mappings with respect to the $L^\infty$ norm by Lemma 1 (resp. by corollary 4.3 in Castro et al. (2021)), so as $k \to \infty$, $U_k \to U_P^\pi$ and $O_k \to U^\pi$. By taking limit on the left hand side of the above inequality, we recover that $U_P^\pi(x,y) \leq U^\pi(x,y)$. $\qquad\square$

