# OpenReview forum: "Revisiting Bisimulation: A Sampling-Based State Similarity Pseudo-metric"
_ICLR.cc/2023/TinyPapers — Submitted to Tiny Papers @ ICLR 2023_

### Official Review · Reviewer_xL93 · 2023-03-20

**Confidence:** 3

**Summary Of Contributions:**

A new metric with on-policy guarantees but need further improvement in writing

**Rating:**

Clear, Correct, and Reproducible (CCR): a submission which meets the reviewing criteria

**Strengths And Weaknesses:**

Summary:

The authors propose a a new (pseudo) metric, inspired by the MICo distance, which is called PMiCo (Pseudo-metric Matching under independent Couplings). This metric is desirable because it can be learnt from transition samples and yields on-policy guarantees.

Strength:

It seems the proposed sampling-based bisimulation metric outperforms the baseline and has on-policy guarantees. The proof in the appendix looks correct.

Weakness:

1. The authors should double-check the notations. For example, when the authors introduce the definition of the bi-simulation metric d~ , the notation X is different from \mathcal{X}. Does X also denote the state space as \mathcal{X} does? And there are two wasserstein distance in the paper, W in d~  and \mathcal{W} in π-bisimulation. Do them denote the same distance?
2. How the bisimulation metric is defined is not clear to me. Without further explanations to fixed point of an operator, it’s hard to understand what is the bisimulation metric indeed. What’s more, how do you confirm the fixed point is unique? Any references?
3. The motivation to propose the new bisimulation metric is not strong enough. I would recommend to add several sentences in the introduction and background section to explain what is sampling-based method and why we need that.
4. Figure 1 need a better legend, it's hard to tell which one is the proposed method and how it shows that PMiCo metric yields a tighter bound.

**Suggested Changes:**

Please take a loot at the weakness

---

> ### Author Response · Authors · 2023-06-01
> **Answers to your questions**
>
> Dear reviewer xL93, thank you for taking the time to review our submission and for your detailed feedback! We hope to address your concerns below.
>
> 1/ We apologize for these typos and thank you for bringing this to our attention. \mathcal{X} refers to the state space while W refers to the Wasserstein distance. We fixed the notations in the new version of our submission.
>
> 2/ The fixed point of an operator is a point which value does not change by application of this operator. More formally, a fixed point of a mapping F on a set X is a point x∈X for which F(x)=x. We included this definition in the paragraph where the bisimulation is introduced in the new version of our submission.The unicity of the fixed point of the bisimulation operator comes from Banach s theorem and is well-know in the literature. Following your suggestion, we added a citation in our submission to clarify this point. Note that the application of Banach s theorem is similar to the one made in the proof of theorem 1 in appendix A.
>
> 3/ In the background, we present three existing distances: the $\pi$-bisimulation metric, the MiCo distance and the projected MiCo distance.
> - the $\pi$-bisimulation metric has a high computational cost and introduces
> a bias under sampled transitions. Sample-based algorithms are at the heart of reinforcement learning which is concerned with learning from experience or interactions. We emphasised this point both in the introduction and the background sections.
> - the MiCo distance overcomes this issue and can be learnt from samples while supporting the lipschitz continuity of the value function. This is important because one of the aims of representation learning in RL is that two states that are close in the feature space are also close in the space of value functions. However, this distance is not a pseudo metric and in particular does not guarantee the self-distance property which is problematic from an implementation perspective.
> - The projected MiCo distance is an alternative used by Castro et al, 2021 to overcome the self-distance issue from which the MiCo distance suffer. It can be though of as a heuristic algorithm as it does not provide any theoretical bound on the value function.
>
> The PMiCo pseudo metric that we introduce can both be learnt from samples and yields strong theoretical guarantees for the value function (see Table 1). We highlighted these points in the new version of the paper, thank you for bringing this up.
>
> 4/ Admittedly, the legend of figure 1 is a little bit hard to parse. We replaced the symbols in the legend by the full name of the distance to improve the clarity, namely the $\pi$-bisimulation metric, the MiCo distance, the projected MiCo distance and the PMiCo pseudo metric which we introduce. Thank you for allowing us to highlight this.
> The value bound gap depicted in figure 1 is tighter for PMiCo than its sample-based counterpart, the MiCo distance, as predicted by our theory (theorem 2).
> While  the bound from the π-bisimulation and projected MICo distances is tighter, unlike PMiCo, the π-bisimulation is biased when approximated using only sampled  transitions, which is the setting at the heart of reinforcement learning, and the projected MICo can be negative and may not upper bound the value differences. On the contrary, the PMiCo supports the lipschitz continuity of the value function and by construction does not suffer from any biases issue in its estimation from sample states. We clarified this point at the end of Section 3.
>
> We hope this addresses your concerns and thank you again for reviewing our paper.
>
> In light of the clarifications above, we kindly invite you to reconsider your assessment.
> In particular, we would like to opt-in for archival.

---

### Official Review · Reviewer_zBJn · 2023-03-30

**Confidence:** 2

**Summary Of Contributions:**

The paper is concerned with finding representations of states in reinforcement learning. The goal is have a metric that approximately  two nearby states are also nearby in the value functions space. The paper proposes a new metric that overcomes certain limitations (theoretical and practical) of prior works

**Rating:**

Great Start (GS): a submission which meets some of the reviewing criteria but has room for improvement

**Strengths And Weaknesses:**

**Strengths**

- The paper states the problem that needs to be solved in a clear manner. This allows even a generalist machine learning researcher to get a sense for the problem being solved
- Empirical evidence is provided to back the theory developed in the paper that clarifies the advantages of proposed approach

**Weaknesses**

- The paper uses the term `pseudo-metric` which is a fairly technical term. Could the authors clarify what is the difference between `pseudo-metric` and a `metric`?
- Please consider using text in the figure caption as opposed to symbol to simplify Figure 1's interpretation
- The paper claims that the proposed metric is lower than `MICo` but two other metrics in Figure 1 are lower. Does this suggest that the other metrics are better for practical use? Some discussion may help with the presentation
- Consider including code that accompanies your work

**Suggested Changes:**

Please check **Weaknessess** for suggested changes

---

> ### Author Response · Authors · 2023-06-01
> **Answering your questions**
>
> Dear reviewer zBJn, thank you for taking the time to review our submission and for your detailed feedback! We hope to address your concerns below.
>
> 1/ A metric $d$ on a set $\mathcal{X}$ is a function $d: \mathcal{X} \times \mathcal{X} \rightarrow [0, \infty[$ respecting the following axioms for any $x, y, z \in \mathcal{X}$:
> - $d(x, y)=0 \Longleftrightarrow x=y$
> - $d(x, y)=d(y, x)$
> - $d(x, y) \leq d(x, z)+d(z, y)$
>
> A pseudometric respects the same axioms except that the first condition above is weakened as follows: $x=y \implies d(x, y)=0$. This means that it is possible to have $d(x, y)=0$ and $x \neq y$.
>
> 2/ Admittedly, the legend of figure 1 is a little bit hard to parse. We replaced the symbols in the legend by the full name of the distance to improve the clarity, namely the $\pi$-bisimulation metric, the MiCo distance, the projected MiCo distance and the PMiCo pseudo metric which we introduce. Thank you for allowing us to highlight this.
>
> 3/ The value bound gap depicted in figure 1 is tighter for PMiCo than its sample-based counterpart, the MiCo distance, as predicted by our theory (theorem 2).
> While  the bound from the π-bisimulation and projected MICo distances is tighter, unlike PMiCo, the π-bisimulation is biased when approximated using only sampled  transitions, which is the setting at the heart of reinforcement learning, and the projected MICo can be negative and may not upper bound the value differences. On the contrary, the PMiCo supports the lipschitz continuity of the value function and by construction does not suffer from any biases issue in its estimation from sample states. We clarified this point at the end of Section 3.
>
> We hope this addresses your concerns and thank you again for reviewing our paper.
>
> In light of the clarifications above, we kindly invite you to reconsider your assessment.
> In particular, we would like to opt-in for archival.

---

### Meta-Review · Area_Chair_Z5PK · 2023-04-07

**Recommendation:** Invite to revise
**Confidence:** 4

**Metareview:**

Quality: The paper is of good quality, as it presents a clear problem and proposes a new metric that has empirical evidence to support its advantages over prior works. The authors provide a solid theoretical framework, and the proofs in the appendix appear to be correct. However, there are some weaknesses in the notations and the definition of the bisimulation metric that could be addressed.

Clarity: The paper is generally well-written and easy to follow, with a clear statement of the problem being solved and an explanation of the proposed solution. The authors provide examples and figures to help illustrate their ideas. However, there are some technical terms and notations that could be clarified, and the legend for Figure 1 could be improved.

Originality: The paper proposes a new metric, PMiCo, which is inspired by the MICo distance but overcomes certain limitations of prior works. The authors provide empirical evidence to support the advantages of their approach, and the metric has on-policy guarantees. Overall, the paper offers a novel contribution to the field of reinforcement learning.

**Summary:**

 the paper presents a clear problem but need some revising

**Comments And Feedback To The Authors:**

1. the authors should clarify the difference between a pseudo-metric and a metric, as it is a technical term that may not be clear to all readers. Additionally, using text in the figure caption instead of symbols may simplify the interpretation of Figure 1.

2. the authors should double-check their notations to ensure consistency, as there seem to be discrepancies between X and \mathcal{X} and two different Wasserstein distances denoted as W and \mathcal{W}. Additionally, the definition of the bisimulation metric is not clear, and further explanations on the fixed point of an operator are needed. The authors should also strengthen the motivation for proposing the new bisimulation metric by providing background information on sampling-based methods and why they are necessary.

3. the authors should provide more discussion on the comparison of PMiCo with the other metrics in Figure 1, as two of the metrics appear to be lower than PMiCo. Additionally, the authors may consider providing code to accompany their work.

Overall, the paper presents a clear problem and proposes a new metric that has empirical evidence to back its advantages over prior works. However, there are some weaknesses that need to be addressed, such as clarifying technical terms and notations, providing more explanation on the bisimulation metric and its fixed point, and strengthening the motivation for proposing the new metric. Additionally, a better legend for Figure 1 and the inclusion of accompanying code would improve the presentation of the paper.

**Reason For Not Giving A Higher Recommendation:**

Notations and symbols used in the paper could be clarified to ensure consistency and avoid confusion. The definition of the bisimulation metric is not clear enough, and further explanations are needed to understand the fixed point of an operator and how to confirm its uniqueness. The motivation for proposing the new bisimulation metric could be strengthened by providing background information on sampling-based methods and why they are necessary. The legend for Figure 1 could be improved to help the reader interpret the results more clearly.

**Reason For Not Giving A Lower Recommendation:**

N/A

---

### Decision · Program_Chairs · 2023-04-08

Revision accepted; invite to archive

---

> ### Author Response · Authors · 2023-06-01
> **Opting-in for archival**
>
> Dear all,
>
> Thank you very much for time reviewing our paper and for your insightful comments. We did our best to incorporate your feedback into the new version of our paper and believed this made our paper stronger.
>
> In light of the clarifications we detail [here](https://openreview.net/forum?id=lkWvTn2IzA&noteId=4ubEjWCf8G0) and [there](https://openreview.net/forum?id=lkWvTn2IzA&noteId=kR2-SNoNUH), we kindly invite you to reconsider your assessment. In particular, we would like to opt-in for archival.

---

> > ### Comment · Area_Chair_Z5PK · 2023-06-07
> > **the paper is ok for archival**
> >
> > The authors have improved the paper a lot and resolved the concerns of reviewers. After checking this paper, I confirmed that this work meets the threshold for archival, contents the URM statement and is deanonymized.